# IL-17A Is the Critical Cytokine for Liver and Spleen Amyloidosis in Inflammatory Skin Disease

**DOI:** 10.3390/ijms23105726

**Published:** 2022-05-20

**Authors:** Shohei Iida, Takehisa Nakanishi, Fumiyasu Momose, Masako Ichishi, Kento Mizutani, Yoshiaki Matsushima, Ai Umaoka, Makoto Kondo, Koji Habe, Yoshifumi Hirokawa, Masatoshi Watanabe, Yoichiro Iwakura, Yoshihiro Miyahara, Yasutomo Imai, Keiichi Yamanaka

**Affiliations:** 1Department of Dermatology, Mie University Graduate School of Medicine, 2-174 Edobashi, Tsu 514-8507, Japan; kmcasters@clin.medic.mie-u.ac.jp (S.I.); t-nakanishi@clin.medic.mie-u.ac.jp (T.N.); k-mizutani@med.mie-u.ac.jp (K.M.); matsushima-y@clin.medic.mie-u.ac.jp (Y.M.); umaokaai@clin.medic.mie-u.ac.jp (A.U.); pjskt886@yahoo.ne.jp (M.K.); habe-k@clin.medic.mie-u.ac.jp (K.H.); imaiyasutomo@gmail.com (Y.I.); 2Department of Personalized Cancer Immunotherapy, Mie University Graduate School of Medicine, 2-174 Edobashi, Tsu 514-8507, Japan; momose-fumi@clin.medic.mie-u.ac.jp (F.M.); miyahr-y@clin.medic.mie-u.ac.jp (Y.M.); 3Department of Oncologic Pathology, Mie University Graduate School of Medicine, 2-174 Edobashi, Tsu 514-8507, Japan; masako-i@doc.medic.mie-u.ac.jp (M.I.); ultray2k@med.mie-u.ac.jp (Y.H.); mawata@doc.medic.mie-u.ac.jp (M.W.); 4Center for Animal Disease Models, Research Institute for Biomedical Sciences, Tokyo University of Science, Chiba 278-8510, Japan; iwakura@rs.tus.ac.jp; 5Imai Dermatology Pediatric Dermatology Allergology, 5-1-1 Ebie, Fukushima, Osaka 553-0001, Japan

**Keywords:** inflammatory skin, mouse model, dermatitis, cytokine, amyloidosis, JAK inhibitor, IL-17A

## Abstract

Systemic amyloidosis is recognized as a serious complication of rheumatoid arthritis or inflammatory bowel disease, but also of inflammatory skin disease. However, the detailed molecular mechanism of amyloidosis associated with cutaneous inflammation remains unclear, and therapeutic approaches are limited. Here, we investigated the pathophysiology of amyloidosis secondary to cutaneous inflammation and the therapeutic effects of Janus kinase (JAK) inhibitors by examining a mouse model of spontaneous dermatitis (KCASP1Tg mice). Moreover, KCASP1Tg mice were crossed with interleukin-17A (IL-17A) knockout mice to generate IL-17A-/KCASP1Tg and examine the role of IL-17A in amyloidosis under cutaneous inflammation. KCASP1Tg mice showed severe amyloid deposition in the liver and spleen. Increased serum-neutral fat levels and decreased lymphocyte production were observed in the spleen. Overproduction of amyloidosis was partially ameliorated by the administration of JAK inhibitors and was further improved in IL-17A-/KCASP1Tg mice. IL-17A-producing cells included CD4, gamma delta, and CD8 T cells. In summary, our results from the analysis of a mouse model of dermatitis revealed that skin-derived inflammatory cytokines can induce amyloid deposition in the liver and spleen, and that the administration of JAK inhibitors and, even more, IL-17A ablation, reduced amyloidosis. This study demonstrates that active control of skin inflammation is essential to prevent internal organ amyloidosis.

## 1. Introduction

Systemic amyloidosis is a rare disorder in which abnormal proteins accumulate in tissues and organs, and is often associated with chronic inflammation. Chronic inflammatory diseases such as rheumatoid arthritis, inflammatory bowel disease, and systemic lupus erythematosus can induce the liver-mediated overproduction of serum amyloid A protein, thus causing secondary amyloid A protein (AA) amyloidosis [1,2].

Psoriasis and atopic dermatitis are chronic and intractable inflammatory skin diseases, and skin inflammation may be associated with amyloid deposition [3,4,5]. Primary localized cutaneous amyloidosis, such as lichen amyloidosis, is characterized by severely itchy, scaly, and reddish patches of thickened skin with multiple small bumps. Because of its benign nature, few studies have examined the detailed molecular mechanism of systemic organ amyloidosis, and therapeutic options are limited.

During the inflammatory process, dendritic cells and T cell-derived cytokines, including type I interferons, tumor necrosis factor-α (TNF-α), IL-17A, and IL-22 stimulate keratinocytes to synthesize proinflammatory cytokines, such as IL-1β, IL-6, TNF-α, IL-23, which can reach the liver where they promote the production of amyloid A [6]. In particular, cutaneous inflammation, such as psoriasis, is characterized by high expression of IL-17A, which acts on immune and non-immune cell types and strongly contributes to tissue inflammation. Overproduction and continuous systemic release of skin-derived inflammatory cytokines may result in distant organ damage, leading to cardiovascular disorders such as cardiac sclerosis and cerebrovascular disease [3,7].

In the current study, we used a spontaneous inflammatory skin model, keratin 14-driven caspase-1 overexpressing KCASP1Tg mice. The KCASP1Tg mice started to show the first symptoms of dermatitis at approximately 8 weeks of age, and the cutaneous inflammation spread from the face to the whole body [8,9,10]. The current dermatitis model reveals dermatitis without any external triggers. The skin overexpresses caspase-1 in the basal layer of keratinocyte, which results in the activation of IL-1β and IL-18. The dermatitis characteristics and histological and behavioral profile fulfil seven out of eight of the Hanifin and Rajka diagnostic criteria for atopic dermatitis, and have been used as a model for the development of atopic dermatitis [8].

It has become a well-known fact that cutaneous inflammation is not only a localized problem of the skin, but also causes inflammation of organs throughout the body, resulting in comorbidities and a shortened life expectancy. One of these complications is amyloidosis. The systemic amyloidosis shares many non-specific symptoms including fatigue, weakness, loss of appetite, and weight loss. Edema of the ankles due to kidney or heart lesions, paresthesia in the limbs due to nerve lesions, and shortness of breath due to heart lesions may also occur [11]. In this model mice, amyloid is produced in large amounts locally in the skin, but this serum amyloid A type 3 remains in the skin and does not predominate in the systemic circulation. On the other hand, inflammatory cytokines produced in the interaction of leukocytes and keratinocytes are carried into the bloodstream and lead to the overproduction of SAA1 and SAA2 in the liver. These will likely be deposited in many distant organs. The deposition of SAA1 and SAA2 has been detected in the gastrointestinal tract, resulting in hypoalbuminemia due to albumin leakage [4]. The amyloid deposition was also detected in the kidneys and spleen, resulting in organ collapse at the late phase [3,12].

In the current study, we showed that IL-17A is an essential cytokine for amyloidosis of the liver and spleen associated with cutaneous inflammation. Because the site of cutaneous inflammation is composed of a network of many different inflammatory cytokines, multi-cytokine-targeted drugs—in this case Janus kinase (JAK) inhibitors—were also tested as a potential therapeutic approach.

## 2. Results

### 2.1. Histological Analysis Showed the Amyloid Deposition in the Liver of KCASP1Tg Mice

With the exacerbation of skin inflammation, the visceral organs were markedly enlarged at the 16-week period. To examine amyloid deposition, livers were collected from KCASP1Tg and WT mice at the age of 16 weeks and stained with hematoxylin and eosin (H&E), direct fast scarlet (DFS), and Congo red. Severe amyloid deposition was detected in the livers of KCASP1Tg mice compared to that of WT mice. Histological examination revealed that amyloid depositions were radiating from the Glisson’s capsule, such as the interlobular bile ducts, interlobular arteries, and interlobular veins, to the central veins (Figure 1a, the positive control of DFS staining was shown in Appendix A). There was a significant increase in the DFS staining positive area in KCASP1Tg mice (Figure 1b).

### 2.2. Histological Analysis of Spleen in KCASP1Tg Mice

We examined whether amyloid deposition, observed in the liver, was also detected in the spleen. At 16 weeks of age, spleens were sampled from KCASP1Tg and WT mice and stained with H&E, DFS, and Congo red. Severe amyloid deposition was detected in the spleen of KCASP1Tg mice compared to that of WT mice. In particular, amyloid depositions were primarily observed in the marginal zone, which is the boundary between the white and red pulps (Figure 2a,b). Immunostaining revealed a decrease in CD4, CD8, and CD20 cells and an increase in CD138-positive cells in KCASP1Tg mice (Figure 2c).

### 2.3. KCASP1Tg Mice Showed Signs of Liver and Spleen Dysfunction

The peripheral lymphocyte count was evaluated and plasma biochemistry was examined at 16 weeks to determine the effect of amyloid deposition on the physiological function of liver and spleen. Serum liver transaminase levels were elevated in KCASP1Tg mice. Neutral fat levels were also significantly elevated in KCASP1Tg mice compared to those in WT mice. The concentrations of cholinesterase and total cholesterol were unchanged between the two groups (Figure 3a). The peripheral blood lymphocyte count was elevated in KCASP1Tg mice compared to that in WT mice (Figure 3b). The CD8 counts were increased in KCASP1Tg mice. The total number of lymphocytes, CD4 T cells, CD8 T cells, and B cell counts were unchanged in the superficial lymph nodes. The counts of total lymphocytes, CD4, CD8, and CD20 in the entire spleen decreased in KCASP1Tg mice. In addition, the counts per weight of spleen (count/g of each spleen) decreased in KCASP1Tg mice.

### 2.4. Effect of Administration of JAK Inhibitors and IL-17A Ablation

To verify whether amyloid deposition is caused by excessive cytokines released by the dermatitis lesions, we investigated the effect of two treatment methods aiming to suppress inflammatory cytokines. The multiple inflammatory cytokine protein expressions including TNF-α, IL-17A, and IL-23 in the skin lesions of KCASP1Tg mice were elevated at 16 weeks compared to week 8 (Appendix A). KCASP1Tg mice were treated daily with either 5 mg/kg baricitinib or 5 mg/kg cerdulatinib, two kinds of JAK inhibitors, for 8 weeks. Histopathological features were ameliorated in KCASP1Tg mice treated with JAK inhibitors. Specifically, in the skin, epidermal hyperplasia, hyperkeratosis, parakeratosis, sponginess, and increased infiltration of mixed inflammatory cells were significantly improved in baricitinib- and cerdulatinib-treated KCASP1Tg mice (Appendix A). In the liver and spleen, DFS-positive and Congo red-positive areas were significantly recovered in those mice. Next, to investigate the effect of IL-17A in dermatitis-related amyloidosis, IL-17A-/KCASP1Tg mice were examined. Amyloidosis in both liver and spleen was ameliorated in IL-17A-/KCASP1Tg mice (Figure 4a,b), and the expression level of SAA in the liver was decreased compared with 16-week-old KCASP1Tg mice (Appendix A). Moreover, the generation of B cell follicles and CD20 + B cells in the marginal zone of the spleen was also recovered in cerdulatinib-treated KCASP1Tg mice and IL-17A-/KCASP1Tg mice (Figure 4b). Severe amyloid staining area was reduced in the liver (Figure 4c left) and spleen (Figure 4c right) by JAK inhibitor treatment or in IL-17A deficient KCASP1Tg mice. There were no differences in the expression of above inflammatory cytokines in the liver and spleen of 0, 8, and 16-week-old KCASP1Tg mice, JAK inhibitor treatment, and IL-17A-/KCASP1Tg mice (data not shown).

### 2.5. IL-17A-Producing Cell in Dermatitis Model Mice

As mentioned above, in KCASP1Tg mice, dermatitis first appears in the face, and cervical lymph nodes are involved by the inflammation of the facial skin. The lymphocytes from cervical lymph nodes were isolated and investigated to identify IL-17-producing cells. Staining for IL-17A and immune cell markers revealed that IL-18 receptor-positive CD4 T cells, gamma-delta T cells, and CD8 T cells were found to be the major IL-17A-producing cells in this order (Figure 5).

## 3. Discussion

In addition to being a barrier against pathogens, the skin is one of the largest immune organs and functions as an alarmin, releasing various inflammatory and pro-inflammatory cytokines in response to external and intrinsic stimuli. Here, we investigated the association between dermatitis and amyloid depositions in the liver and spleen by examining a transgenic mouse model of cutaneous inflammation, KCASP1Tg mice.

From previous study, we know that inflammation of the skin affects the arteries, and that arteriosclerosis is detected not only in the abdominal aorta but also in the peripheral basilar arteries [7]. These arterial abnormalities were partially improved by the administration of antibody preparations [7]. Cytokines produced locally in the skin may reach the adipose tissue of the abdomen through the bloodstream, leading to the burning of adipocytes and the release of adipocytokines, which contribute to the systemic inflammatory cascade [13]. Based on these facts, the effects on arteries may be due to direct effects on the vascular endothelium by sustained elevated inflammatory cytokines in the blood [14], as well as the direct effects of adipocytokines from the fatty tissues surrounding the major blood vessels [13]. In fact, statistics have shown that patients with psoriasis vulgaris, atopic dermatitis, and eczema show a high complication rate of coronary artery disease and cerebrovascular disease, which are often fatal [15,16,17,18]. The concept of the hardening of this vasculature is expressed in the term “inflammatory skin march” [14]. In the inflammatory skin condition, it has also been reported that osteoporosis may be complicated due to a decrease in the vascular network and the increase in osteoclasts coupled with the decrease in osteoblasts [19]. Male infertility is also related to sperm hypoplasia, presumably caused by the direct effect of an increase in inflammatory cytokines from skin lesions [20].

Persistent refractory dermatitis can lead to organ amyloidosis [3,4,5]. A large amount of cytokines released from the involved area of persistent inflammation may be mixed in the bloodstream, causing the liver to produce amyloid A protein. Serum amyloid A protein (SAA) belongs to a family of apolipoproteins that are constitutively produced in several organs. SAA is also an acute-phase protein produced in response to or enhanced by inflammatory stimulation, including proinflammatory cytokines IL-1, IL-6, and TNF-α, especially in the liver [21,22]. Acute phase SAA has several functions, such as the recruitment of immune cells to the inflammation sites and transport of cholesterol to the liver. Among the SAAs, SAA3 is produced in the skin, and although SAA3 is highly concentrated in the skin, its concentration in the blood did not increase in inflammatory mice, while SAA1 and SAA2 mainly produced in the liver are predominantly increased in inflammatory mouse plasma and are deposited in organs [4].

In the current study, we demonstrated that mice with skin eruptions developed liver and spleen amyloidosis. Severe amyloid deposition was detected in the liver of KCASP1Tg mice, and dense deposition was detected radiating from Glisson’s capsule around the interlobular bile ducts, interlobular arteries, and interlobular veins to the central veins. This was probably due to the blood flow from Glisson’s capsule to the central veins.

Severe amyloid deposition was also observed in the spleens of KCASP1Tg mice compared to those of WT mice. Serious deposition was observed mainly in the marginal zone, which is the boundary between the white and red pulps. A closer look at the spleen showed that the characteristics of resident cells in the spleen and CD4-, CD8-, and CD20-positive cells decreased in KCASP1Tg mice, especially around the lymph follicle. The marginal zone is where a unique population of B cells called marginal zone B cells is rich [23], and amyloid deposition results in a decrease in the number of B cells. Since the marginal zone region also serves as a developmental site for immature B cells that have just arrived from the bone marrow, changes in the structural microenvironment of the spleen caused by amyloid deposition may affect B cell maturation, B-T interactions, and differentiation into plasma cells. Further disintegration may lead to immunosuppression owing to the difficulty in the maturation of lymphocytes in the spleen. In contrast, CD138-positive plasma cells were enriched in KCASP1Tg mice. Plasma cells are terminally differentiated secretory cells that play a critical role in humoral immunity by producing copious amounts of soluble antibodies, low-affinity IgM, and secreting high-affinity IgG, IgA, or IgE [24,25]. High concentrations of IgG and IgE have been reported in the current dermatitis model [8,9,10,12]. High levels of inflammatory cytokines can support plasma cell survival, likely enabling these cells to survive the period of inflammation [26]. A decrease in CD20-positive B cells and increase in CD138-positive plasma cells are opposing events. An increase in specific cytokines may promote B cell activation and differentiation into plasma cells but reduce the number of unstimulated naïve B cells in the spleen of KCASP1Tg mice.

We then investigated the systemic consequences of liver and spleen dysfunction. With regard to liver function, mildly elevated transaminase and neutral fat levels were observed. In current models, disruption of organ function has been reported over a long period of time [3], but at 4 months of age, the changes were mild. This abnormality is consistent with the findings of human liver amyloidosis. In chronic inflammatory diseases, such as rheumatoid arthritis, inflammatory bowel disease, and autoimmune diseases, mild increase in liver enzymes is often observed, and amyloid deposition may be actually progressing in such situation.

An increase in the lymphocyte count was detected in the peripheral blood, similar to that observed in inflammatory conditions. The CD8 count was also significantly increased in KCASP1Tg mice. The absolute numbers of lymphocytes and each component were unchanged in the superficial lymph nodes between the two groups. However, in the spleen, the counts of total lymphocytes, CD4, CD8, and CD20 decreased in KCASP1Tg mice. The counts per weight of the spleen also decreased in KCASP1Tg mice, suggesting impaired productivity in the spleen. The weight of the spleen was approximately four times larger in the inflamed mice than in the normal mice, compensating for the loss of function. Further disintegration may lead to immunosuppression due to difficulty in leukocyte maturation. The lymph nodes of inflamed mice also showed an increase in size to compensate for the maturation of lymphocytes.

Inflammatory cytokines, including TNF-α and IL-6, are produced in chronic inflammatory sites, which may migrate into the liver, leading to the production of amyloid A. Recently, a multi-cytokine-targeted drug, JAK inhibitor, has been clinically used in the treatment of psoriasis and atopic dermatitis, with a good response. Here, using a mouse model of spontaneous dermatitis, we examined its effect on amyloidosis by suppressing multiple cytokines using JAK inhibitors. Baricitinib is a selective JAK1 and JAK2 inhibitor, while cerdulatinib hydrochloride is an orally active, multi-targeted tyrosine kinase inhibitor, especially for JAK1, JAK2, JAK3, and TYK2. Skin eruptions were ameliorated in JAK-treated mice, and amyloidosis was significantly reduced in cerdulatinib-treated KCASP1Tg mice.

Although mice responded to JAK inhibitors, IL-17A inhibition alone in the IL-17A-/KCASP1Tg mice was more effective in significantly reducing the amyloid deposition. IL-17A signaling is independent of the JAK pathway. The discordance in results, in which JAK is not involved in IL-17A signaling but has a similar inhibitory effect, may be due to the fact that other inflammatory cytokines mediating the JAK pathway are suppressed by JAK inhibitor treatment, thereby controlling inflammation itself, and in the process reducing IL-17A expression, which in turn controls amyloid deposition.

The major IL-17-producing cells were the IL-18R-positive T cells, suggesting that IL-18 production from injury or inflammation in keratinocytes may trigger IL-17 production. It has been reported that T cells secrete IL-17A in response to IL-18 without antigen [27]. Surgical samples of healthy breast or abdominal skin stimulated with IL-17 produced SAA [28]. In contrast, there are reports showing that SAA is also involved in IL-17-producing cell differentiation [29], thus supporting an inflammatory positive loop between IL-17 and SAA.

Taken together, our data show that persistent dermatitis can cause liver and spleen amyloidosis, affecting the organ functionality and immune features. Although JAK inhibitors and IL-17 ablation improved amyloidosis, active control of dermatitis is recommended.

## 4. Materials and Methods

### 4.1. Animals

Eight-week-old female transgenic mice, in which human caspase-1 gene is expressed under the keratinocyte-specific keratin 14 promoter (KCASP1Tg) [7], and C57BL/6N littermate (WT) mice were used. KCASP1Tg was also crossed with IL-17A knockout mice (IL-17A-/KCASP1Tg mouse) [8]. The mice were housed in an environmentally-conditioned room at 21 ± 2 °C, with 12:12 h light cycle, 60% humidity, and food and water available ad libitum. All mice were sacrificed and analyzed at 16 weeks of age. The experimental protocol was approved by the Mie University Board Committee for Animal Care and Use (No.22-39-5-1).

### 4.2. Blood Sampling and Clinical Chemistry Parameters

All mice were euthanized with CO_2_. Blood was sampled from the tail vein or by cardiac puncture, placed in a 1.5 mL tube containing heparin (MOCHIDA PHARMACEUTICAL CO., LTD. Tokyo, Japan), and centrifuged (6000 rpm for 5 min) to separate the plasma. The collected plasma was stored at −80 °C until examination. The concentrations of serum liver transaminase, cholinesterase, neutral fat, and total cholesterol were measured using a Hydrasys 2 (Sebia, Lisses, France).

### 4.3. Lymphocytes Staining in the Blood, Spleen, and Lymph Nodes

To stain lymphocyte surface markers, freshly collected whole blood was first incubated with ACK Lysing Buffer (Thermo Fisher Scientific, Waltham, MA, USA) to lyse erythrocytes, and then directly stained with monoclonal antibodies, CD4-PE, CD8-APC, or CD20-FITC antibody (BD Biosciences, Franklin Lakes, NJ, USA) in cell surface staining buffer containing 0.1 M phosphate-buffered saline and 2% FCS (Biowest, Nuaillé, France). The spleen and superficial lymphocytes were also sampled by gentle grinding between sterile ground glass slides, incubated with ACK lysing buffer, and directly stained with monoclonal antibodies, CD4-PE, CD8-APC, or CD20-FITC antibody. The total counts of lymphocytes, CD4 + T cells, CD8 + T cells, and CD20 + B cells were analyzed using a BD Accuri C6 flow cytometer (BD Biosciences, Franklin Lakes, NJ, USA).

### 4.4. Amyloid and Infiltrating Cells Staining

The livers and spleens were collected and fixed in 10% neutral buffered formalin solution (Wako, Osaka, Japan). Tissue sections at 5 µm thickness were obtained. The H&E, DFS, and Congo red staining were performed on the livers and spleens, and the stained sections were analyzed using ImageJ JS (https://imagej.nih.gov/ij/index.html, accessed on 27 March 2022). The percentage of DFS staining positive lesions was calculated. To identify the specific type of infiltrating cells in the spleen, CD138 (Thermo Fisher Scientific, Waltham, MA, USA), CD4, CD8, and CD20 (Cell Signaling Technology, Danvers, MA, USA) staining was performed.

### 4.5. Cytokine and SAA Analysis

The dorsal skin and liver tissues were smashed using a pestle after being frozen with liquid nitrogen. Subsequently, these tissues were homogenized with 5 mL of RIPA Buffer with Protease Inhibitor Cocktail (NACALAI TESQUE INC., Kyoto, Japan) per gram of tissue. The lysates of dorsal skin and liver tissues were sonicated and centrifuged at 10,000× *g* for 15 min at 4 °C, and the supernatant collected. The skin and liver cytokine expressions, such as TNF-α, IL-22, IL-23, and IL-17A, were assayed by enzyme-linked immunosorbent assay (ELISA) according to the manufacturer’s instructions. Similarly, the expression level of SAA in the liver was also assayed (TNF-α, IL-22, IL-23, and SAA; R&D Systems, Minneapolis, MN, USA, IL-17A; Medix Biochemica, Besançon, France).

### 4.6. Oral Administration of JAK Inhibitors

Eight-week-old female KCASP1Tg and WT littermate mice were orally treated with the selective JAK1 and JAK2 inhibitor baricitinib (OYC1, Oriental Yeast, Kyoto, Japan) or an orally active multi-targeted tyrosine kinase inhibitor, especially JAK1, JAK2, JAK3, and TYK2 cerdulatinib hydrochloride (Astellas, Tokyo, Japan). The treatment schedule was as follows: 5 mg/kg baricitinib or 5 mg/kg cerdulatinib was administered every day, as previously reported [4,9]. All mice were sacrificed at 16 weeks of age, and the livers and spleens were analyzed.

### 4.7. IL-17A-Producing Lymphatic Primary Cells

Cells from the cervical lymph nodes were incubated in culture medium for 4 h in the presence of monensin and phorbol 12-myristate 13-acetate (Sapphire Bioscience Pty. Ltd. Redfern NSW, Australia)/ionomycin (Wako). Cells were first stained with IL-18 receptor (IL-18R)-FITC (Miltenyi Biotec, Gaithersburg, MD, USA), CD8s-PerCp-Cy5.5, gamma delta TCR-APC, and BioLegend Brilliant Violet 421 anti-mouse CD4 antibody (BioLegend, San Diego, CA, USA). Dead cells were eliminated using a LIVE/DEAD Fixable Aqua Dead Cell Stain Kit (Thermo Fisher Scientific). Cells were fixed with 4% (*w*/*v*) paraformaldehyde (nacalai tesque), permeabilized with 0.1% saponin buffer (phosphate-buffered saline with 0.1% saponin, 1 mM HEPES; Thermo Fisher Scientific, and 0.1% bovine serum albumin; Sigma-Aldrich, St. Louis, MO, USA), and then stained with anti-IL-17A or a control rat IgG1 antibody (BD Biosciences), as previously described [10]. The cells were examined using a FACS Canto II (BD Biosciences), and the data were analyzed using FlowJo software (v10.5) (Tree Star, Ashland, OR, USA).

### 4.8. Statistical Analysis

Statistical analyses were performed using the PRISM software version 9 (GraphPad, San Diego, CA, USA). Two-group comparisons were analyzed using the Mann–Whitney test, and more than three groups were analyzed using the Mann–Whitney test or ordinary one-way ANOVA. Differences were considered statistically significant at *p* < 0.05. *; *p* < 0.05, **; *p* < 0.01, ***; *p* < 0.001, ****; *p* < 0.0001.

## Figures and Tables

**Figure 1 ijms-23-05726-f001:**
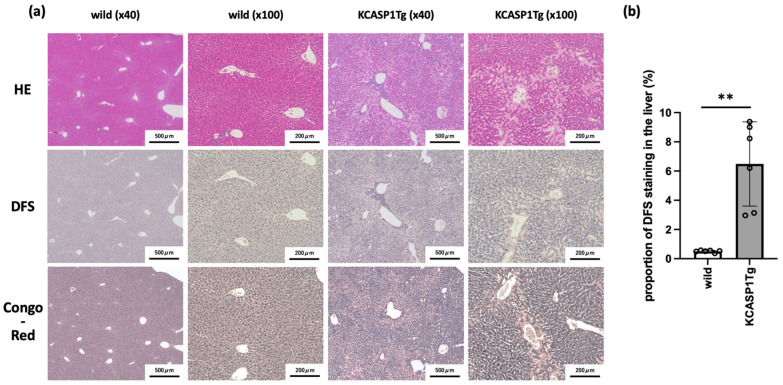
Histological analysis showed the amyloid deposition in the liver of KCASP1Tg mice. The liver was collected from 16-week-old KCASP1Tg (*n* = 6) and WT (*n* = 6) mice and stained with hematoxylin and eosin (H&E), direct fast scarlet (DFS), and Congo red. (**a**) Severe amyloid deposition was detected in the livers of KCASP1Tg mice compared to in WT mice. In particular, there were massive deposits radiating from the Glisson’s capsule, including the interlobular bile ducts, interlobular arteries, and interlobular veins, to the central veins of the liver. Images at ×40 and ×100 magnification are shown. The positive control of DFS staining is supplemented in Appendix A. (**b**) DFS staining-positive area is quantified by Mann–Whitney test, and there was a significant increase in KCASP1Tg mice. The data is expressed as mean ± SD. ** *p* = 0.0022 between wild-type and KCASP1Tg mice.

**Figure 2 ijms-23-05726-f002:**
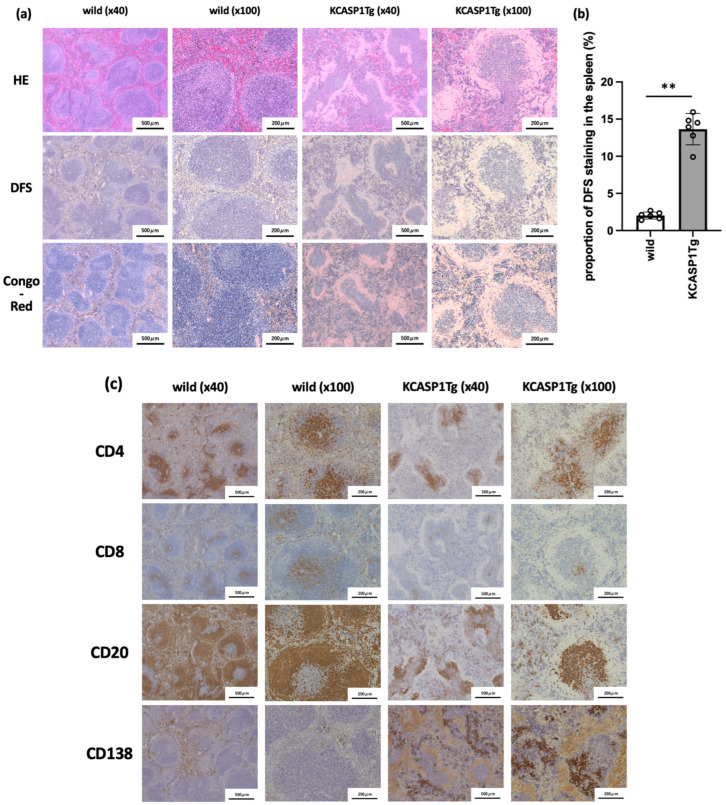
Histological analysis of spleen in KCASP1Tg mice. At 16 weeks of age, the spleen was sampled from KCASP1Tg and WT mice (*n* = 6 per group) and analyzed. H&E, DFS, and Congo red staining were performed. (**a**) H&E, DFS, and Congo-red. Massive amyloid deposition was detected in KCASP1Tg mice. Marginal zone is the boundary between the white and red pulp, and dense deposition was detected. (**b**) DFS staining-positive area is quantified by Mann–Whitney test, and there was a significant increase in DFS positive area in KCASP1Tg mice. The data is expressed as mean ± SD. ** *p* = 0.0022 between wild-type and KCASP1Tg mice. (**c**) Immunostaining for CD4, CD8, CD20, and CD138 was performed and CD4, CD8, and CD20-positive cells were apparently decreased in KCASP1Tg mice. On the contrary, CD138-positive cells were more stained in KCASP1Tg mice.

**Figure 3 ijms-23-05726-f003:**
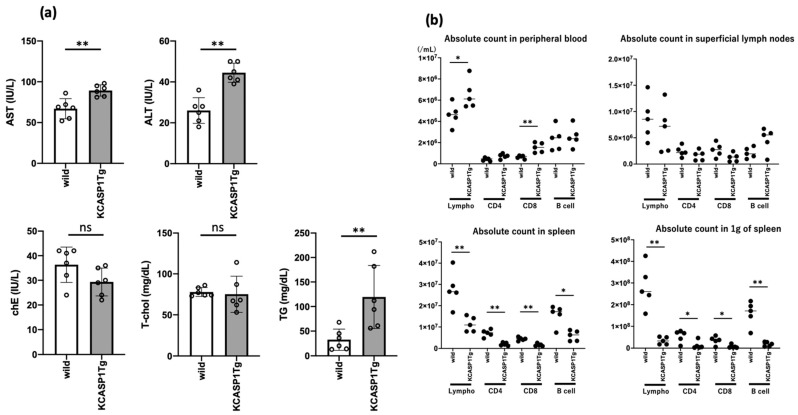
KCASP1Tg mice showed elevated serum liver transaminase and neutral fat level and insufficient production of lymphocytes. Plasma biochemical analysis and lymphocyte count in both 16-week-old KCASP1Tg and WT mice were performed (*n* = 6 per group). (**a**) The concentration of serum liver transaminase, neutral fat levels were elevated in KCASP1Tg mice with significance (*p* values; AST, 0.0087; ALT, 0.0022; TG, 0.0087, respectively). (**b**) Peripheral blood lymphocyte count was elevated in KCASP1Tg mice compared to WT (*p* = 0.0317). The amounts of CD8 were also increased in KCASP1Tg mice (*p* = 0.0079). In superficial lymph nodes, the absolute number of lymphocytes, CD4 T cell, CD8 T cell, and B cell counts were unchanged. However, the counts of total lymphocyte, CD4, CD8, and CD20 in the whole spleen were decreased in KCASP1Tg mice. The counts per weight of spleen were also decreased in KCASP1Tg mice. Values are presented as mean ± SD, and statistically significant differences (* *p* < 0.05, ** *p* < 0.01) are indicated. (AST, aspartate aminotransferase; ALT, alanine aminotransferase; chE, cholinesterase; T-chol, total cholesterol; TG, triglyceride; ns, not significant).

**Figure 4 ijms-23-05726-f004:**
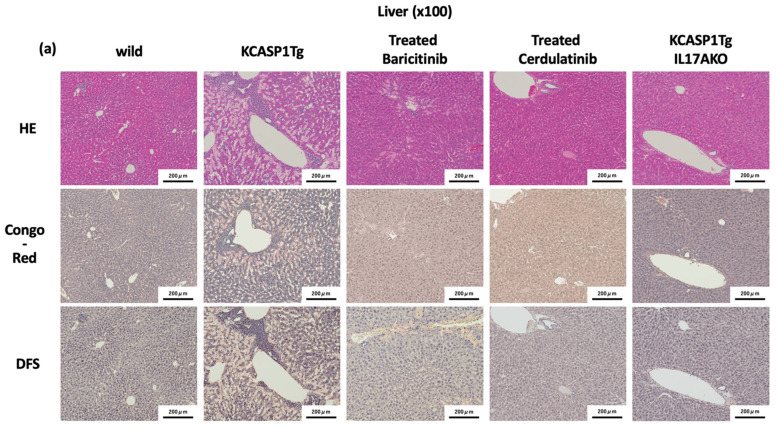
Effect of administration of JAK inhibitors and IL-17A ablation. KCASP1Tg mice were treated with baricitinib or cerdulatinib from 8 to 16 weeks of age (Treated Baricitinib and Treated Cerdulatinib, respectively; *n* = 6 per group). Histopathological investigation was also performed in IL-17A-/KCASP1Tg mice (*n* = 6). (**a**,**b**) Histopathological analysis of treated and untreated KCASP1Tg was performed. Amyloid deposition in the liver was recovered by baricitinib administration and further recovered in cerdulatinib-treated KCASP1Tg mice. IL-17 deletion rescued the amyloidosis dramatically. Similarly in the spleen, massive deposition in the marginal zone is reduced in cerdulatinib-treated KCASP1Tg mice, and rescued in IL-17A-/KCASP1Tg mice. In particular, CD20-positive cells are recovered by these treatments. Images at ×100 magnification are shown. (**c**) DFS-positive areas were recovered in baricitinib-and cerdulatinib-treated KCASP1Tg mice and in IL-17A-/KCASP1Tg mice (left; liver, right; spleen). All data are expressed as the mean±SD. * *p* < 0.05, **** *p* < 0.0001 compared to wild mice by ordinary one-way ANOVA test.

**Figure 5 ijms-23-05726-f005:**
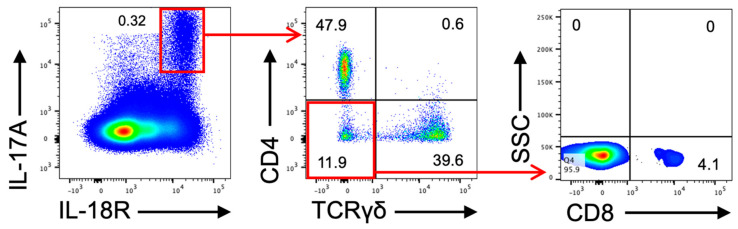
Identification of IL-17A-producing cell in KCASP1Tg mice. Cells from the cervical lymph nodes were isolated and analyzed via flow cytometry to identify IL-17-producing lymphocytes. The numbers indicate the percentage of cells in each gate. Note that IL-18R-positive CD4 T cells, gamma delta T cells, and CD8 T cells were found to be the major IL-17-producing cells in this order. (SSC, side scatter).

## Data Availability

Not applicable.

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
