# Peer review of "IL-17A Is the Critical Cytokine for Liver and Spleen Amyloidosis in Inflammatory Skin Disease"

_ijms, 2022, doi:10.3390/ijms23105726_

Round 1

Reviewer 1 Report

The study by Keiichi Yamanaka and the team describes IL-17A is a key cytokine for liver and spleen amyloidosis in inflammatory skin disease.  They have evaluated the hypothesis that “inflammatory cytokines from chronic dermatitis can also affect amyloid deposition in the distant organs, including the liver and spleen” using the KCASP1Tg mice modal.

The study is interesting and important facts could be acquired by this study about the relationship and interplay of skin-derived inflammatory cytokines on chronic dermatitis, and amyloid deposition in internal organs.

The study is excellent in general, and the manuscript is well-written.

Relevant details are given in the introduction and have identified areas that need further investigation

Results are clearly presented with the description

The discussion is well written supporting the results, and compared with outcomes of previous investigations.

Materials and Methods are written briefly but give necessary details to reproduce the experiments.

The references list looks ok

This manuscript does not appear to have many flaws. Hence, I suggest a minor review.

My comments to improve your manuscript are as follows,

  1. Abstract line 29 – provide the explanation of the abbreviation IL-17A as “interleukin (IL)-17A” since it is the first time.
  2. Figure 3 – what are the abbreviations AST, ALT, chE, T-chol means? I guess they are the Serum liver transaminase, cholinesterase, neutral fat, and total cholesterol. But it is difficult to understand.
  3. Did you carry out three independent determinations for each of these experiments to confirm the repeatability of the outcomes? If so please mention that in each figure caption.
  4. Materials and Methods – line 302 – provide the specifications of the instrument you used “automated analyzer”
  5. Materials and Methods – section 4.3 - line 303 – provide relevant references to the method

Author Response

Responses to the comments of Reviewer #1

The study by Keiichi Yamanaka and the team describes IL-17A is a key cytokine for liver and spleen amyloidosis in inflammatory skin disease. They have evaluated the hypothesis that “inflammatory cytokines from chronic dermatitis can also affect amyloid deposition in the distant organs, including the liver and spleen” using the KCASP1Tg mice modal. The study is interesting and important facts could be acquired by this study about the relationship and interplay of skin-derived inflammatory cytokines on chronic dermatitis, and amyloid deposition in internal organs. The study is excellent in general, and the manuscript is well-written. Relevant details are given in the introduction and have identified areas that need further investigation

Results are clearly presented with the description. The discussion is well written supporting the results, and compared with outcomes of previous investigations. Materials and Methods are written briefly but give necessary details to reproduce the experiments. The references list looks ok. This manuscript does not appear to have many flaws. Hence, I suggest a minor review.

My comments to improve your manuscript are as follows,

  1. Abstract line 29 – provide the explanation of the abbreviation IL-17A as “interleukin (IL)-17A” since it is the first time.

Response: Thank you for your suggestion. This has been corrected.

  1. Figure 3 – what are the abbreviations AST, ALT, chE, T-chol means? I guess they are the Serum liver transaminase, cholinesterase, neutral fat, and total cholesterol. But it is difficult to understand.

Response: We have listed the official name in the figure legend. We appreciated for your suggestion.

  1. Did you carry out three independent determinations for each of these experiments to confirm the repeatability of the outcomes? If so please mention that in each figure caption.

Response: For each experiment, several preliminary experiments were conducted during the preparation phase, and the actual data collection was performed by sampling a number of n=6. This has been mentioned in each figure legend.

  1. Materials and Methods [1]– line 302 – provide the specifications of the instrument you used “automated analyzer”

Response: This has been supplemented.

  1. Materials and Methods – section 4.3 - line 303 – provide relevant references to the method

Response: Thank you for your suggestion. This staining method was performed on mice using the same technique used in routine clinical examinations and is not based on any specific reference.

Reviewer 2 Report

The authors hypothesized that inflammatory cytokines from chronic dermatitis could also affect amyloid deposition in distant organs, including the liver and spleen. They reported that the spontaneous dermatitis model of KCASP1Tg mice revealed that skin-derived inflammatory cytokines could induce amyloid deposition in the liver and spleen, and administration of JAK inhibitors and, even more, IL-17A ablation reduced amyloidosis. I expected a study about the relationship between skin-derived inflammatory cytokines and systemic inflammation; however, the author only focuses on the liver and spleen. From my point of view, I would suggest providing the following results first to demonstrate that the skin-derived inflammatory cytokines can indeed affect amyloid deposition in distant organs first:

  1. The skin histopathological and cytokines expressions such as L-17A, IL-23, IL-22, and TNF-αresults of KCASP1Tg mice on Day0, 8, and 16weeks to support that 8 and 16 weeks are the appropriate time point for further studies.
  2. Then, demonstrate the difference in cytokines protein expression between skin, spleen, and liver, and the amyloid deposition at 8 and 16 weeks.

Other major issues:

  1. There is no information about the amyloid deposition in distant organs in the introduction section. The author must improve it.
  2. The author introduced the Th1/Th17 and Th2 related inflammatory skin diseases in the introduction section but lacked information about the model of chronic inflammatory of KCASP1Tg mice. Is that model belong to Th1/Th17 or Th2 related inflammatory skin diseases?
  3. The author should also provide the results of skin histopathological and cytokine expressionsto support the JAK inhibitors diminishing skin dermatitis and inflammation status.

Minor issues:

  1. The positive staining of DFS is not apparent.
  2. The scale bar is not easy to read.

Based on my experience, I agree that IL-17A plays an essential role in the Th1/Th17 related inflammatory skin disease; however, the rationale, the study design, and the results of this study can not convince me. I would encourage the author to improve the above issues first, re-edit the manuscript, and then re-submit it.

Author Response

Responses to the comments of Reviewer #2

The authors hypothesized that inflammatory cytokines from chronic dermatitis could also affect amyloid deposition in distant organs, including the liver and spleen. They reported that the spontaneous dermatitis model of KCASP1Tg mice revealed that skin-derived inflammatory cytokines could induce amyloid deposition in the liver and spleen, and administration of JAK inhibitors and, even more, IL-17A ablation reduced amyloidosis. I expected a study about the relationship between skin-derived inflammatory cytokines and systemic inflammation; however, the author only focuses on the liver and spleen. From my point of view, I would suggest providing the following results first to demonstrate that the skin-derived inflammatory cytokines can indeed affect amyloid deposition in distant organs first:

  1. The skin histopathological and cytokines expressions such as L-17A, IL-23, IL-22, and TNF-α results of KCASP1Tg mice on Day0, 8, and 16weeks to support that 8 and 16 weeks are the appropriate time point for further studies.
  2. Then, demonstrate the difference in cytokines protein expression between skin, spleen, and liver, and the amyloid deposition at 8 and 16 weeks.

Response: Thank you for your great suggestion. First of all, this mouse model of dermatitis reveals spontaneous dermatitis without any external triggers. The skin overexpresses capase-1, which results in the activation of IL-1β and IL-18. The dermatitis characteristics and histological and behavioral profile fulfil 7 out of 8 of the Hanifin and Rajka diagnostic criteria for atopic dermatitis, and have been used as a model of atopic dermatitis, as described in the following paper (J Immunol. 2000 Jul 15;165(2):997-1003, Nat Immunol. 2000 Aug;1(2):132-7, Proc Natl Acad Sci U S A. 2002 Aug 20;99(17):11340-5, and several papers). However, as reviewer suspected, there is no pure type 2 disease similar to human atopic dermatitis, and of course, among the interaction between infiltrating lymphocytes and monocytes and skin tissue, there are many blood cells that produce type 1 cytokines and type 3 cytokines, and this is the state of so-called "contamination".

It has become a well-known fact that cutaneous inflammation is not only a localized problem of the skin, but also causes inflammation of organs throughout the body, resulting in comorbidities and a shortened life expectancy. One of these complications is amyloidosis, as mentioned in the discussion, amyloid is produced in large amounts locally in the skin, but this serum amyloid A type 3 remains in the skin and does not predominate in the systemic circulation. On the other hand, inflammatory cytokines produced in the interaction of leukocytes and keratinocytes triggered by dermatitis, are carried into the bloodstream and lead to the overproduction of SAA1 and ASS2 in the liver. These will likely be deposited in many distant organs. As indicated in the text, we have reported the deposition of SAA1 and ASS2 in the gastrointestinal tract, resulting in hypoalbuminemia due to albumin leakage (Int J Mol Sci. 2021 Dec 21;23(1):28.). We have also shown that deposits in the kidneys and spleen can cause organ disorganization (J Dermatol Sci. 2017 Oct;88(1):146-148, PLoS One. 2014 Aug 13;9(8):e104479.). This article focuses on the liver and spleen, detailing how amyloidosis causes dysfunction and effects on lymphocyte synthesis. This is the first paper to prove in vivo that IL-17A plays a very important role in the production of amyloid, while pan-JAK also works. We are also seeking submissions for a special issue of this journal on the possibility that increased inflammatory cytokines produced by dermatitis may cause various organ changes. This paper is one of the contributions.

As the reviewer 2 pointed out, we examined the protein expression of cytokines, TNF-α, IL-22, IL-23, and IL-17A in the skin, liver, and spleen of 0, 8, 16-week-old KCASP1Tg mice. In addition, the production of SAA in the liver of JAK inhibitor-treated, IL-17A-/KCASP1Tg mice at 16-week-old, and wild mice at 16-week-old were also examined. KCASP1Tg mice showed a predominant increase in L-17A, IL-23, and TNF-α from week 8 to 16 in the skin. On the other hand, no difference was observed in IL-22, and no significant differences in the expression of these cytokines was observed in the liver and spleen. In SAA, there was a similar predominant increase from KCASP1Tg mice from 8 to 16 weeks, with no predominant difference in the JAK inhibitor-treated, but a prominent decrease in the IL-17A-/KCASP1Tg mice; the lack of significant difference in the JAK inhibitor-treated was thought to be due to the small number of individuals. The administration of a JAK inhibitor from 8 weeks is desirable. The above explanation and the additional data have been supplemented in the revised version. After consulting with many immunologists, we have come to the conclusion that it is not possible to search for subtle changes in cytokine-producing cells by immunostaining, so we do not do this. This matter has been reported to the editorial team. We appreciated for your great suggestion.

Other major issues:

  1. There is no information about the amyloid deposition in distant organs in the introduction section. The author must improve it.

Response: We have supplemented the explanation of amyloid deposition in the several organs in the introduction session.

  1. The author introduced the Th1/Th17 and Th2 related inflammatory skin diseases in the introduction section but lacked information about the model of chronic inflammatory of KCASP1Tg mice. Is that model belong to Th1/Th17 or Th2 related inflammatory skin diseases?

Response: Thank you for your great suggestion. As mentioned above, type 2 cytokine producing cells may be predominant in the early phase, similar to human atopic dermatitis; however, type 1 cytokines and type 3 cytokines producing cells are also contaminated in the late phase. We appreciated for your suggestion.

  1. The author should also provide the results of skin histopathological and cytokine expressions to support the JAK inhibitors diminishing skin dermatitis and inflammation status.

Response: Thank you for your suggestion. The pathological findings of chronic dermatitis such as epidermal hyperplasia, hyperkeratosis, parakeratosis, spongiosis, and increased infiltration of mixed inflammatory cells were improved in the JAK inhibitors-treated group in hematoxylin and eosin staining. We have included it as a supplemental figure. We appreciated for your suggestion.  

Minor issues:

  1. The positive staining of DFS is not apparent.

Response: We have supplemented the picture of positive control for DFS in the supplemental figure 1.

.

  1. The scale bar is not easy to read.

 Response: Both the scale bar and the numbers have been replaced.

Based on my experience, I agree that IL-17A plays an essential role in the Th1/Th17 related inflammatory skin disease; however, the rationale, the study design, and the results of this study can not convince me. I would encourage the author to improve the above issues first, re-edit the manuscript, and then re-submit it.

Response: We have supplemented and changed the manuscript.

Round 2

Reviewer 2 Report

The manuscript has significantly improved, and I highly recommend this article be published in IJMS.